

# Effects of six weeks of sub-plateau cold environment training on physical functioning and athletic ability in elite parallel giant slalom snowboard athletes

Tao Ma[1], Jingwang Tan[2], Ran Li[1], Jiatao Li[2] and Binghong Gao[1]

[1] School of Elite Sport, Shanghai University of Sport, Shanghai, Shanghai, China
[2] Department of Sport and Exercise Science, College of Education, Zhejiang University, Hangzhou, Zhejiang, China

Corresponding author
Binghong Gao,
binghong.gao@hotmail.com

## ABSTRACT

**Background**. Hypoxic and cold environments have been shown to improve the function and performance of athletes. However, it is unclear whether the combination of subalpine conditions and cold temperatures may have a greater effect. The present study aims to investigate the effects of 6 weeks of training in a sub-plateau cold environment on the physical function and athletic ability of elite parallel giant slalom snowboard athletes.

**Methods**. Nine elite athletes (four males and five females) participated in the study. The athletes underwent 6 weeks of high intensity ski-specific technical training (150 min/session, six times/week) and medium-intensity physical training (120 min/session, six times/week) prior to the Beijing 2021 Winter Olympic Games test competition. The physiological and biochemical parameters were collected from elbow venous blood samples after each 2-week session to assess the athletes' physical functional status. The athletes' athletic ability was evaluated by measuring their maximal oxygen uptake, Wingate 30 s anaerobic capacity, 30 m sprint run, and race performance. Measurements were taken before and after participating in the training program for six weeks. The repeated measure ANOVA was used to test the overall differences of blood physiological and biochemical indicators. For indicators with significant time main effects, *post-hoc* tests were conducted using the least significant difference (LSD) method. The paired-samples t-test was used to analyze changes in athletic ability indicators before and after training.

**Results**. (1) There was a significant overall time effect for red blood cells (RBC) and white blood cells (WBC) in males; there was also a significant effect on the percentage of lymphocytes (LY%), serum testosterone (T), and testosterone to cortisol ratio (T/C) in females ($p < 0.001 - 0.015$, $\eta_p^2 = 0.81 - 0.99$). In addition, a significant time effect was also found for blood urea(BU), serum creatine kinase (CK), and serum cortisol levels in both male and female athletes ($p = 0.001 - 0.029$, $\eta_p^2 = 0.52 - 0.95$). (2) BU and CK levels in males and LY% in females were all significantly higher at week 6 ($p = 0.001 - 0.038$), while WBC in males was significantly lower ($p = 0.030$). T and T/C were significantly lower in females at week 2 compared to pre-training ($p = 0.007$, $0.008$, respectively), while cortisol (C) was significantly higher in males and females at weeks 2 and 4 ($p_{(male)} = 0.015$, $0.004$, respectively; $p_{(female)} = 0.024$, $0.030$, respectively). (3) There was a noticeable increase in relative maximal oxygen uptake, Wingate 30 s

relative average anaerobic power, 30 m sprint run performance, and race performance in comparison to the pre-training measurements ($p < 0.001 - 0.027$).

**Conclusions**. Six weeks of sub-plateau cold environment training may improve physical functioning and promote aerobic and anaerobic capacity for parallel giant slalom snowboard athletes. Furthermore, male athletes had a greater improvement of physical functioning and athletic ability when trained in sub-plateau cold environments.

## INTRODUCTION

Generally, training in the hypoxic environment of a plateau is more effective than training on a plain, when using the same training load under both conditions (*Gao et al., 2019*; *Park et al., 2020*). Previous studies have confirmed that altitude or hypoxic training can increase athletes' blood volume and hemoglobin (Hb) levels (*Bonne et al., 2014*; *Tannheimer et al., 2010*; *Wachsmuth et al., 2013*), improve the body's ability to buffer acids such as blood lactic acid (BLA) (*Ramos-Campo et al., 2018*; *Shaw et al., 2020*), and increase the number of capillaries and mitochondria in skeletal muscle (*Hoppeler et al., 2003*; *Vogt et al., 2001*). At present, studies concerning the effects of plateau or hypoxic training on athletes' physical functional status and athletic ability have mainly focused on physical events that occur during the summer, such as rowing, swimming, cycling, middle-distance running, boxing, and sparring (*Gao, Gao & Meng, 2018*; *Garcia et al., 2020*; *Millet & Brocherie, 2020*). Athletes in these events have improved their physical functional and athletic ability through plateau or hypoxic training and have achieved excellent athletic results in major competitions at home and abroad (*Arezzolo et al., 2020*; *Gao et al., 2019*; *Gao, Gao & Meng, 2018*; *Viscor et al., 2018*; *Zhao, Li & Lu, 2008*).

The snowboarding parallel giant slalom is a competitive event with a single race time of about 35–45 s. The sport utilizes athletes' glycolytic metabolism for energy, and is most often trained for and competed in on high altitude mountains (*Li, Chen & He, 2018*; *Zebrowska et al., 2012*). Normally, the training environment for the snowboarding parallel giant slalom is extremely harsh and athletes are subjected to cold and lack of oxygen at high altitudes. Compared to the plains environment, the athletes' cardiopulmonary regulation (*Saltykova, 2016*; *Zafeiratou et al., 2021*), thermoregulation (*Castellani et al., 2010*; *Castellani & Young, 2016*), energy metabolism (*Lichtenbelt et al., 2014*; *Marino, Sockler & Fry, 1998*), immune system (*LaVoy, McFarlin & Simpson, 2011*), and neuroendocrine system (*Cui et al., 2014*; *Makinen et al., 2008*) are significantly different in hypoxic and cold environments. It has been shown that when exercising in cold environments, convective heat dissipation increases due to the thermal gradient of temperature conduction and energy expenditure increases. The body's heat production and dissipation become imbalanced under these conditions, resulting in a drop in body temperature, increased muscle viscosity, and reduced physical functional status (*Luo, 2005*; *Weng, Wang & Lin, 2019*). Some studies

have reported that athletes in winter sports experience greater heat and water loss during exhalation, with a water loss of 32 mg/L during exercise at temperatures below −16 °C, proportional to the amount of ventilation (*Sue-Chu, 2012*). Exercise in cold environments also causes the endocrine and immune systems to respond, leading to immunosuppression (*Cicchella, Stefanelli & Massaro, 2021*; *Mourtzoukou & Falagas, 2007*). Prolonged and repeated cold exposure also increases the activity of the neuroendocrine system, releasing more neurotransmitters and causing reduced sympathetic activation and increased parasympathetic activation (*Dan et al., 2002*; *Makinen et al., 2008*; *Marino, Sockler & Fry, 1998*). Therefore, based on the respective effects of hypoxia and cold, it can be assumed that the physiological responses of the body systems in combined settings will be more acute and complex.

At present, research on the status of physical functioning and athletic ability in winter sports is mainly related to cross-country skiing, biathlon, Nordic biathlon, and other long-distance physical events. There is less research on the physical functional status of athletes in skill-led or speed-led snowboarding sports such as the snowboarding parallel giant slalom, snowboarding U-course tricks, snowboarding slopestyle tricks, snowboarding big jump, and snowboarding obstacle chase. Numerous studies have found that altitude training significantly increases the red blood cell and hemoglobin levels of cross-country skiers and biathletes, which, in turn, promotes higher maximum oxygen uptake and improved performance (*Bădău et al., 2016*; *Christoulas, Karamouzis & Mandroukas, 2011*; *Sandbakk & Holmberg, 2017*; *Wehrlin, Marti & Hallen, 2016*). In view of the specificity of the oxygen-deficient cold environment of the altitude in winter, it is not known how these conditions affect the physical functional status and athletic ability of athletes. Hence, this study aims to investigate the effects of 6 weeks of training in a sub-plateau cold environment on the physical functional and athletic ability of elite snowboarding parallel giant slalom athletes. The results of the current study may provide theoretical and practical guidance for other snowboarding sub-sports and other winter skill-dominated speed sports.

# MATERIALS AND METHODS

## Participants and ethical principles

The study required all athletes participating in this trial to be in good health and free from contraindications, such as cardiovascular disease, that would make them unfit for competitive training. Nine of the 12 elite snowboarding parallel giant slalom national team athletes were found to meet the inclusion criteria. These nine athletes did not suffer any injuries during the training and testing process and all athletes completed all of the training and testing in this study under the supervision of their coaches. The study was conducted in accordance with the Declaration of Helsinki and was approved by the Ethics Committee of the Shanghai University of Sport (ethics number: 102772020127082). All athletes understood the content and process of the test and signed an informed consent form. Demographic information about the subjects is shown in Table 1.

**Table 1  Basic athlete information.**

| Gender | Age (years) | Height (cm) | Weight (kg) | Training experience (years) |
|--------|-------------|-------------|-------------|-----------------------------|
| male   | 26.75 ± 4.03 | 178.75 ± 3.50 | 73.63 ± 2.87 | 12.25 ± 3.86 |
| female | 26.40 ± 3.85 | 168.80 ± 8.98 | 61.40 ± 4.93 | 14.00 ± 4.53 |

Notes.
  Values are presented as mean ± SD.

## Training arrangements

This training program was conducted six weeks before the Olympic Test Competition held in Chongli District (altitude: 1,878–2,031 m), Hebei Province, China. The athletes were trained six days per week (Monday to Saturday) from 9:00 am to 11:30 am in a cold, outdoor, sub-plateau environment (average temperature: −10 °C ∼−20 °C) with high intensity training in specialized skiing techniques. Each athlete made eight to 10 ski trips per morning (approximately 560 m/trip). In the afternoon (15:00 to 17:00), the athletes underwent medium-to-high intensity physical training at 1,500 m above sea level. The physical training programs consisted of low-to-medium intensity dynamic stretching, aerobic training, medium-to-high intensity explosive training in upper and lower body, and stability training. The training was followed by a conditioning break on Saturday afternoon and Sunday. The Polar Team Pro team heart rate monitoring system (Polar Team Pro, Finland) was used to monitor the intensity of the training load during the indoor physical training sessions and the outdoor ski-specific technical training sessions. The Polar Team Pro team heart rate monitoring system works by transmitting each athlete's heart rate in real time to the coach's iPad, who then provides verbal feedback to the athlete on their heart rate information. The weekly schedule for the specialized skiing techniques training and physical training is shown in Table 2.

## Blood index test and procedure

Elbow venous blood was collected from athletes at 7:00 a.m. at the following timepoints: pre-training, week 2, week 4, and post-training. The blood samples were collected from the elbow vein using an ethylene diamine tetraacetic acid anti-coagulated vacuum blood collection tube for white blood cells (WBC), the percentage of lymphocytes (LY%), the percentage of mononuclear cells (MO%), the percentage of neutrophils (NE%), red blood cells (RBC), and hemoglobin (Hb) testing (Beckman Coulter AC Tdiff-2 hemocytometer; Beckman Coulter, San Jose, CA, USA). A total of 2.5 ml of elbow vein blood was collected using a sodium heparin anti-coagulated vacuum blood collection tube for analyzing blood urea (BU), creatine kinase (CK), testosterone (T), and cortisol (C) (Beckman Coulter Access 2 Immunosav System; Beckman Coulter, San Jose, CA, USA) (*Elegańczyk-Kot et al., 2011*).

## Athletic ability assessment

### 30 m sprint run test

A 30 m sprint run test was performed before and two days after the training period. The 30 m sprint test was measured by a dedicated coach by using a hand stopwatch (Casio S053

Ma et al. (2023), *PeerJ*, DOI 10.7717/peerj.14770

**Table 2** The schedule of 6-week training program of elite snowboarding parallel giant slalom athletes in the sub-plateau cold environment prior to the Olympic Test competition.

| Week | Specialized skiing techniques training | | Physical training | | | | | | | |
|---|---|---|---|---|---|---|---|---|---|---|
| | | | Aerobic warm-up | | Dynamic stretching | | Strength training | | Aerobic running | |
| Week 1 | Dose: 560 m/trip, 8 trips/time, 6 times/week | Intensity: High (85%–90%HR$_{max}$) | Dose: 15 min/time, 6 times/week | Intensity: Low (60%–70%HR$_{max}$) | Dose: 15 min/time, 6 times/week | Intensity: Low (60%–70%HR$_{max}$) | Dose: 45 min/time, 6 times/week | Intensity: High (85%–90%HR$_{max}$) | Dose: 15 min/time, 6 times/week | Intensity: Low (60%–70%HR$_{max}$) |
| Week 2 | Dose: 560 m/trip, 8 trips/time, 6 times/week | Intensity: High (85%–90%HR$_{max}$) | Dose: 15 min/time, 6 times/week | Intensity: Low (60%–70%HR$_{max}$) | Dose: 15 min/time, 6 times/week | Intensity: Low (60%–70%HR$_{max}$) | Dose: 60 min/time, 6 times/week | Intensity: High (85%–90%HR$_{max}$) | Dose: 15 min/time, 6 times/week | Intensity: Low (60%–70%HR$_{max}$) |
| Week 3 | Dose: 560 m/trip, 10 trips/time, 6 times/week | Intensity: High (85%–90%HR$_{max}$) | Dose: 15 min/time, 6 times/week | Intensity: Low (60%–70%HR$_{max}$) | Dose: 15 min/time, 6 times/week | Intensity: Low (60%–70%HR$_{max}$) | Dose: 60 min/time, 6 times/week | Intensity: High (85%–90%HR$_{max}$) | Dose: 15 min/time, 6 times/week | Intensity: Low (60%–70%HR$_{max}$) |
| Week 4 | Dose: 560 m/trip, 10 trips/time, 6 times/week | Intensity: High (85%–90%HR$_{max}$) | Dose: 15 min/time, 6 times/week | Intensity: Low (60%–70%HR$_{max}$) | Dose: 15 min/time, 6 times/week | Intensity: Low (60%–70%HR$_{max}$) | Dose: 60 min/time, 6 times/week | Intensity: High (85%–90%HR$_{max}$) | Dose: 15 min/time, 6 times/week | Intensity: Low (60%–70%HR$_{max}$) |
| Week 5 | Dose: 560 m/trip, 8 trips/time, 5 times/week | Intensity: High (85%–90%HR$_{max}$) | Dose: 15 min/time, 6 times/week | Intensity: Low (60%–70%HR$_{max}$) | Dose:15 min/time, 6 times/week | Intensity: Low (60%–70%HR$_{max}$) | Dose: 50 min/time, 6 times/week | Intensity: High (85%–90%HR$_{max}$) | Dose: 15 min/time, 6 times/week | Intensity: Low (60%–70%HR$_{max}$) |
| Week 6 | Dose: 560 m/trip, 5 trips/time, 5 times/week | Intensity: Highest (95%–100%HR$_{max}$) | Dose:15min/time, 6 times/week | Intensity: Low (60%–70%HR$_{max}$) | Dose: 25 min/time, 6 times/week | Intensity: Low (60%–70%HR$_{max}$) | Dose: 50 min/time, 6 times/week | Intensity:Medium (70%–85%HR$_{max}$) | Dose: 15 min/time, 6 times/week | Intensity: Low (60%–70%HR$_{max}$) |

**Notes.**

Jogging was the main aerobic warm-up. Dynamic stretching mainly includes toe walk, heel walk, hands-on-knees walk, forward lunge walk, lateral lunge walk, side flexion lunge walk, single leg standing forward flexion, front lunge with turn, back lunge with turn, and caterpillar crawl walk. Aerobic warm-up strength training included Swiss ball back bridge, Swiss ball bide bridge, Swiss ball tuck, prone back raise, supine leg raise, supine two-head raise, alternating right and left box jump, prone push pull, dumbbell lunge squat, dumbbell single leg squat, dumbbell row, explosive grip pull, barbell squat, barbell grip pull, lunge jerk, side toss medicine ball, weighted back raise, Bosu ball single leg squat. Aerobic running training was mainly treadmill jogging.

HF-70W-1DF; Casio Computer Co., Ltd., Tokyo, Japan) to record the time. In the test, all athletes had a 15-min warm-up session and then got ready for the 30 m sprint run test. During the formal test, athletes stood behind the starting line in a standing start position. Athletes were given the command to "run", and all athletes ran across the 30 m finish line as fast as possible. The time taken for each athlete was recorded by a hand stopwatch. Each athlete was tested twice with a five minutes interval and the fastest time was taken as the final test result.

### Wingate 30-second anaerobic athletic ability test

The Wingate 30-second anaerobic athletic ability test was conducted 10 min after the 30 m sprint run test. Before the test, the athletes warmed up on a power bike (Monark Ergomedic 894E, Sweden) with a 60 W load for 10 min and then rode as hard as they could for 30 s. The protocol for the test has been reported in other studies (*Legaz-Arrese et al., 2011*). During the test, the computer automatically recorded the athletes' peak cycling power and relative peak cycling power, which was used as an indicator of the anaerobic capacity. The same procedure was conducted again two days after the six weeks of sub-plateau cold environment training.

### Maximal oxygen uptake assessment

The athlete's maximum oxygen uptake was assessed in the morning, 3 days prior to winter sub-plateau training. The maximal oxygen uptake test was performed on the power bike (Lode Corival Cpet, Groningen, Netherlands) using an incremental load test. An exercise cardiorespiratory testing system (Cortex Metamax 3b, Leipzig, Germany) was used to measure the maximum oxygen uptake of athletes. The exercise cardiorespiratory testing system consisted of a gas analyzer, breathing mask, heart rate monitor, and data analysis software. It uses a mixed gas test method and a breath-per-breath test method to collect real-time data on oxygen consumption, carbon dioxide excretion, respiratory rate, heart rate, respiratory exchange rate, and ventilation during exercise, enabling accurate measurement of the athlete's maximum oxygen uptake. Prior to the test, the athletes put on breathing masks and then warmed up on a power bike for 10 min at a load of 60 W. The formal test was conducted with an initial load of 90 W for the males and 60 W for the females. The riding speed was maintained at 50–60 rpm and then the load was enhanced by 30 W every 2 min. The test was terminated upon self-determined exhaustion or when the participant could no longer maintain the 50-rpm cadence (*Maeda, 2017*). The same procedure was conducted again three days after the 6 weeks of sub-plateau cold environment training.

### Race performance

A snowboarding parallel giant slalom competition was held in the morning, one day prior to sub-plateau cold training. The athlete's time was measured using the Brower Timing Systems (Race Link System, Draper, UT, USA), a professional wireless timing system for ski racing that uses a wireless connection. The system consists of a start beam induction gate, a finish beam induction gate, and a time reception display. When an athlete leaves the start beam sensor gate and crosses the finish beam sensor gate, the race time is immediately

transmitted and displayed on the time receiving display. Before the formal competition, each athlete performed two warm-up skis. In the official test, all athletes stood in the front of the start beam induction gate and crossed the finish beam induction gate as fast as they could. The time taken by each athlete was recorded as the final result. On the fifth day after the 6 weeks of sub-plateau cold environment training, the athletes were tested again in the same way.

## Statistical analysis

All data were processed and analyzed using IBM SPSS Statistics 26.0 software (SPSS Inc., Chicago, USA). Data normality was verified using the Shapiro–Wilk test. The Shapiro–Wilk test showed that all athletes' blood physiological and biochemical parameters and athletic ability data were normally distributed ($p > 0.05$). Repeated measures ANOVA was used to test the overall differences of blood physiological and biochemical indicators during the six weeks of sub-plateau cold environment training. *Post hoc* pairwise comparisons of differences between different time points of the same physiological and biochemical index were performed using the least significant difference (LSD) method. A paired-samples *t*-test was used to analyze the variability of athletic ability indicators before and after training in a sub-plateau cold environment. An independent samples *t*-test was used to compare the differences between groups for the same indicators. Effects sizes (ES) in the form of partial eta squared ($\eta_p^2$) were used from ANOVA output (*Singh et al., 2022*). The magnitude of effects for $\eta_p^2$ was interpreted as small ($<0.06$), moderate ($\geq 0.06-0.13$), and large ($\geq 0.14$) (*Cohen, 1992*). Data are presented as means and standard deviations (mean $\pm$ SD), and $p < 0.05$ was considered to be a significant difference and $p < 0.01$ was considered to be a very significant difference.

## RESULTS

### Physical functional status
#### RBC and Hb

As shown in Table 3, there was a significant time effect in the RBC for males throughout the six weeks of training in the sub-plateau cold environment ($p < 0.001$, $\eta_p^2 = 0.99$), while no significant changes were found in the Hb of both sexes and the RBC of female athletes ($p = 0.096$, $0.377$, $0.685$, respectively; $\eta_p^2 = 0.99$, $0.22$, $0.11$, respectively). The RBC of males was only higher at week 4 ($p = 0.05$), compared to pre-training measurements, while there was no significant difference at week 2 and 6 ($p = 0.632$, $0.774$, respectively). In addition, the RBC and Hb of females were significantly lower than level of males at week 2, 4, and 6 (all $p_{week\ 2} = 0.001$; $p_{week\ 4} = 0.001$, $0.003$, respectively; all $p_{week\ 6} < 0.001$).

#### WBCs and WBC subpopulation cells

Table 4 shows the significant overall time effect for male's WBC and female's LY% throughout the six weeks of training in the sub-plateau cold environment ($p = 0.005$, $0.015$, respectively; $\eta_p^2 = 0.95$, $0.81$, respectively). No significant changes were seen in LY% for male and WBC for female ($p = 0.489$, $0.511$, respectively; $\eta_p^2 = 0.23$, $0.17$, respectively). Similarly, the insignificance was also found in MO% and NE% for both

**Table 3  The changes of RBC and Hb in peripheral blood during 6 weeks sub-plateau cold environment training.**

| Weeks | RBC ($\times 10^{12}$/L) | | Hb (g/L) | |
|---|---|---|---|---|
| | Male ($n = 4$) | Female ($n = 5$) | Male ($n = 4$) | Female ($n = 5$) |
| Pre-training | $5.42 \pm 0.12$ | $4.55 \pm 0.33$ | $167.00 \pm 5.89$ | $144.20 \pm 7.95$ |
| Week 2 | $5.48 \pm 0.14$ | $4.59 \pm 0.21$[**] | $167.50 \pm 5.07$ | $146.00 \pm 6.67$[**] |
| Week 4 | $5.54 \pm 0.12$[a] | $4.61 \pm 0.25$[**] | $168.75 \pm 4.86$ | $147.40 \pm 8.41$[**] |
| Week 6 | $5.46 \pm 0.09$ | $4.49 \pm 0.16$[**] | $167.50 \pm 4.20$ | $142.80 \pm 5.76$[**] |
| Overall effect of $p$-value | <0.001 | 0.685 | 0.096 | 0.377 |
| ES ($\eta_p^2$) | 0.99 | 0.11 | 0.99 | 0.22 |

Notes.
RBC, red blood cells; Hb, hemoglobin; ES, effects sizes.
[a]Significant changes compared to pre-training ($p < 0.05$).
Two asterisks (**) indicate very significant changes compared to male ($p < 0.01$).

**Table 4  The changes in peripheral blood WBC and WBC subpopulations during 6 weeks sub-plateau cold environment training.**

| Weeks | WBC ($\times 10^9$/L) | | LY (%) | | MO (%) | | NE (%) | |
|---|---|---|---|---|---|---|---|---|
| | Male ($n = 4$) | Female ($n = 5$) | Male ($n = 4$) | Female ($n = 5$) | Male ($n = 4$) | Female ($n = 5$) | Male ($n = 4$) | Female ($n = 5$) |
| Pre-training | $5.96 \pm 0.81$ | $6.41 \pm 2.38$ | $47.58 \pm 4.71$ | $32.39 \pm 7.81$ | $5.18 \pm 0.78$ | $5.66 \pm 0.65$ | $45.33 \pm 4.31$ | $52.92 \pm 10.55$ |
| Week 2 | $4.84 \pm 0.19$[a] | $6.34 \pm 1.81$ | $44.43 \pm 8.48$ | $37.40 \pm 5.65$ | $6.05 \pm 1.82$ | $5.66 \pm 0.94$ | $47.40 \pm 6.73$ | $53.68 \pm 5.46$ |
| Week 4 | $5.39 \pm 0.53$ | $6.42 \pm 2.15$ | $46.68 \pm 6.32$ | $40.22 \pm 5.51$[a] | $5.10 \pm 0.29$ | $5.34 \pm 0.93$ | $45.98 \pm 5.91$ | $51.48 \pm 7.45$ |
| Week 6 | $4.98 \pm 0.29$[a] | $5.80 \pm 1.27$ | $48.53 \pm 4.52$ | $47.02 \pm 2.61$[ab] | $5.08 \pm 0.55$ | $5.46 \pm 0.72$ | $44.70 \pm 5.05$ | $51.46 \pm 4.80$[*] |
| Overall effect of $p$-value | 0.005 | 0.511 | 0.489 | 0.015 | 0.575 | 0.145 | 0.703 | 0.920 |
| ES ($\eta_p^2$) | 0.95 | 0.17 | 0.23 | 0.81 | 0.19 | 0.90 | 0.14 | 0.04 |

Notes.
WBC, white blood cell; LY (%), the percentage of lymphocytes; MO (%), the percentage of mononuclear cells; NE (%), the percentage of neutrophils; ES, effects sizes.
[a]Significant changes compared to pre-training ($p < 0.05$).
[b]Significant changes compared to week 2 ($p < 0.05$).
An asterisk (*) indicates significant changes compared to male ($p < 0.05$).

sexes ($p_{male} = 0.575, 0.703, \eta_p^2 = 0.19, 0.14; p_{female} = 0.145, 0.920, \eta_p^2 = 0.90, 0.04$). Compared to the pre-training, the WBC for males decreased significantly at week 2 and week 6 ($p = 0.021, 0.030$, respectively), while there was no significant difference at week 4 ($p = 0.420$). Female LY% increased throughout the training period, with LY% increasing significantly at week 4 and 6 in comparison to level of pre-training ($p = 0.030, 0.038$, respectively), and significantly higher at week 6 compared to week 2 ($p = 0.001$). As for the difference between genders, the NE% for females was significantly higher than that of the males at week 6 ($p = 0.007$), while no significant differences were observed between groups for other indicators ($p = 0.073 - 0.688$).

### BU and CK
Table 5 shows that there was a significant time effect on BU and CK of both males and females throughout the 6 weeks of training ($p_{male} = 0.013, 0.029, \eta_p^2 = 0.68, 0.52; p_{female} = 0.006, 0.001, \eta_p^2 = 0.94, 0.95$). In the paired comparisons within the group, CK
**Table 5 The changes of BU and CK in peripheral blood during 6 weeks sub-plateau cold environment training.**

| Weeks | BU (mmol/L) | | CK (u/L) | |
|---|---|---|---|---|
| | Male ($n = 4$) | Female ($n = 5$) | Male ($n = 4$) | Female ($n = 5$) |
| Pre-training | $5.27 \pm 0.74$ | $4.17 \pm 0.36$ | $111.75 \pm 4.64$ | $84.60 \pm 6.45$ |
| Week 2 | $5.10 \pm 0.92$ | $4.32 \pm 0.33$ | $108.50 \pm 12.29$ | $72.00 \pm 6.83$[a] |
| Week 4 | $5.40 \pm 1.02$ | $4.17 \pm 0.27$[*] | $115.75 \pm 13.56$ | $97.00 \pm 30.47$ |
| Week 6 | $6.02 \pm 1.02$[abc] | $5.67 \pm 0.35$[abc] | $141.75 \pm 20.53$[abc] | $113.20 \pm 12.98$[abc] |
| Overall effect of $p$-value | 0.013 | 0.029 | 0.006 | 0.001 |
| ES ($\eta_p^2$) | 0.68 | 0.52 | 0.94 | 0.95 |

**Notes.**

BU, blood urea; CK, creatine kinase; ES, effects sizes.

[a]Significant changes compared to pre-training ($p < 0.05$).

[b]Significant changes compared to week 2 ($p < 0.05$).

[c]Significant changes compared to week 4 ($p < 0.05$).

An asterisk (*) indicates significant changes compared to male ($p < 0.05$).

was significantly lower for females at week 2 compared to pre-training ($p = 0.023$). BU and CK levels were significantly higher at week 6 in both males and females, compared to pre-training, week 2, and week 4 ($p_{Bu(male)} = 0.001$, 0.001, 0.012, respectively; $p_{CK(male)} = 0.017$, 0.015, 0.003, respectively; $p_{Bu(female)} = 0.007$, 0.009, 0.003, respectively; $p_{CK(female)} = 0.009$, 0.003, 0.043, respectively). However, the BU and CK for males and females were not significantly different at other time points within the group ($p = 0.136 - 1.000$). BU was significantly lower in females than in males at week 4 ($p = 0.049$), and CK was significantly lower in female at week 2 and 6 ($p = 0.029$, 0.026, respectively), revealing further differences between the genders. However, there were no significant differences in BU and CK for females compared to males at other time points ($p = 0.203 - 0.624$).

### T, C, and T/C

It is clear from Table 6 that there was a significant temporal effect on C for both male and female athletes, as well as T and T/C for females throughout the 6 weeks of training in the sub-plateau cold environment ($p_{C(male)} = 0.002$, $\eta_p^2 = 0.98$; $p_{C(female)} = 0.001$, $\eta_p^2 = 0.98$; $p_{T(female)} < 0.001$, $\eta_p^2 = 0.98$; $p_{T/C(female)} = 0.001$, $\eta_p^2 = 0.96$). However, no significant changes were seen in T and T/C for males ($p_{T(male)} = 0.741$, $\eta_p^2 = 0.12$; $p_{T/C(male)} = 0.667$, $\eta_p^2 = 0.13$). Compared to pre-training, C was higher throughout the training for both males and females, and was significantly higher at weeks 2 and 4 ($p_{(male)} = 0.015$, 0.004, respectively; $p_{(female)} = 0.024$, 0.030, respectively). However, no significant difference was observed at week 6 ($p = 0.155 - 0.823$). In addition, the T and T/C for females decreased significantly at week 2 compared to the pre-training measurements ($p = 0.007$, 0.008, respectively), then gradually increased at weeks 4 and 6 ($p = 0.369 - 0.629$), and were significantly higher at week 6 than week 2 ($p < 0.001$, $p = 0.021$, respectively). Furthermore, for comparisons between the gender, both T and T/C were significantly lower in females than in males throughout the training period in the sub-plateau cold environment (all $p_{week2} = 0.002$; all $p_{week4} < 0.001$; $p_{week6} = 0.009$, 0.002, respectively).

**Table 6 The changes of T, C, and T/C in peripheral blood during 6 weeks sub-plateau cold environment training.**

| Weeks | T (ng/dl) | | C (ug/dl) | | T/C | |
|---|---|---|---|---|---|---|
| | Male ($n = 4$) | Female ($n = 5$) | Male ($n = 4$) | Female ($n = 5$) | Male ($n = 4$) | Female ($n = 5$) |
| Pre-training | 753.75 ± 252.34 | 42.20 ± 2.69[**] | 15.97 ± 1.56 | 17.41 ± 1.29 | 48.72 ± 19.25 | 2.48 ± 0.27[**] |
| Week 2 | 707.00 ± 132.43 | 33.40 ± 2.89[a**] | 17.35 ± 1.44[a] | 20.02 ± 1.67[a] | 41.09 ± 7.12 | 1.70 ± 0.19[a**] |
| Week 4 | 764.50 ± 222.11 | 38.80 ± 6.11[**] | 17.63 ± 1.05[a] | 19.40 ± 0.89[a] | 43.50 ± 12.53 | 1.96 ± 0.23[**] |
| Week 6 | 753.25 ± 235.21 | 41.40 ± 3.70[b**] | 16.47 ± 2.37 | 19.89 ± 4.70 | 47.90 ± 21.65 | 2.24 ± 0.43[b**] |
| Overall effect of $p$-value | 0.741 | 0.000 | 0.002 | 0.001 | 0.667 | 0.001 |
| ES ($\eta_p^2$) | 0.12 | 0.98 | 0.98 | 0.98 | 0.13 | 0.96 |

Notes.

T, testosterone; C, cortisol; T/C, testosterone/cortisol; ES, effects sizes.

[a]Significant changes compared to pre-training ($p < 0.05$).

[b]Significant changes compared to week 2 ($p < 0.05$).

An asterisk (*) indicates significant changes compared to male ($p < 0.05$).

Two asterisks (**) indicate very significant changes compared to male ($p < 0.01$).

## Athletic ability

Figure 1 shows that after 6 weeks of intervention, the relative maximal oxygen uptake (Fig. 1A) and snow race performance (Fig. 1D) of both males and females, as well as the 30 m sprint performance (Fig. 1C) of females were considerably higher than their pre-training levels ($p = 0.000 - 0.007$). In addition, Wingate 30 s relative mean power for both male and female athletes (Fig. 1B) and 30 m sprint performance for male subjects (Fig. 1C) were both noticeably higher than pre-training ($p = 0.013 - 0.027$). As for comparisons between gender, the male athletes were very significantly higher than the female athletes in all indicators of athletic ability ($p = 0.000 - 0.001$).

## DISCUSSION

This study was the first to monitor blood physiological and biochemical parameters and the athletic ability of snowboarding parallel slalom athletes during training in a sub-plateau cold environment over a 6-week period. The aim of this study was to investigate the effects of sub-plateau cold environment training on the physical functional status and athletic ability of snowboarding parallel slalom athletes. This study shows that after 6 weeks of high intensity specialized ski technique training and medium-to-high intensity physical training in the sub-plateau cold environment, the elite parallel giant slalom athletes maintained a good functional state of their body systems, in general. The study also found that 6 weeks of sub-plateau hypoxic cold environment training not only improved the physical function of the elite parallel giant slalom athletes, but also promoted significant improvements in specific athletic ability and competition performance. Furthermore, this study found that training in the sub-plateau cold environment was more effective in improving physical functioning and athletic ability in male athletes *versus* female athletes.

### Effects of 6 weeks of sub-plateau cold environment training on the oxygen transport system of elite parallel giant slalom athletes

Snowboarding parallel giant slalom has a competition single trip skiing time of approximately about 35–45 s. Athletes who participate in this sport rely on glycolytic

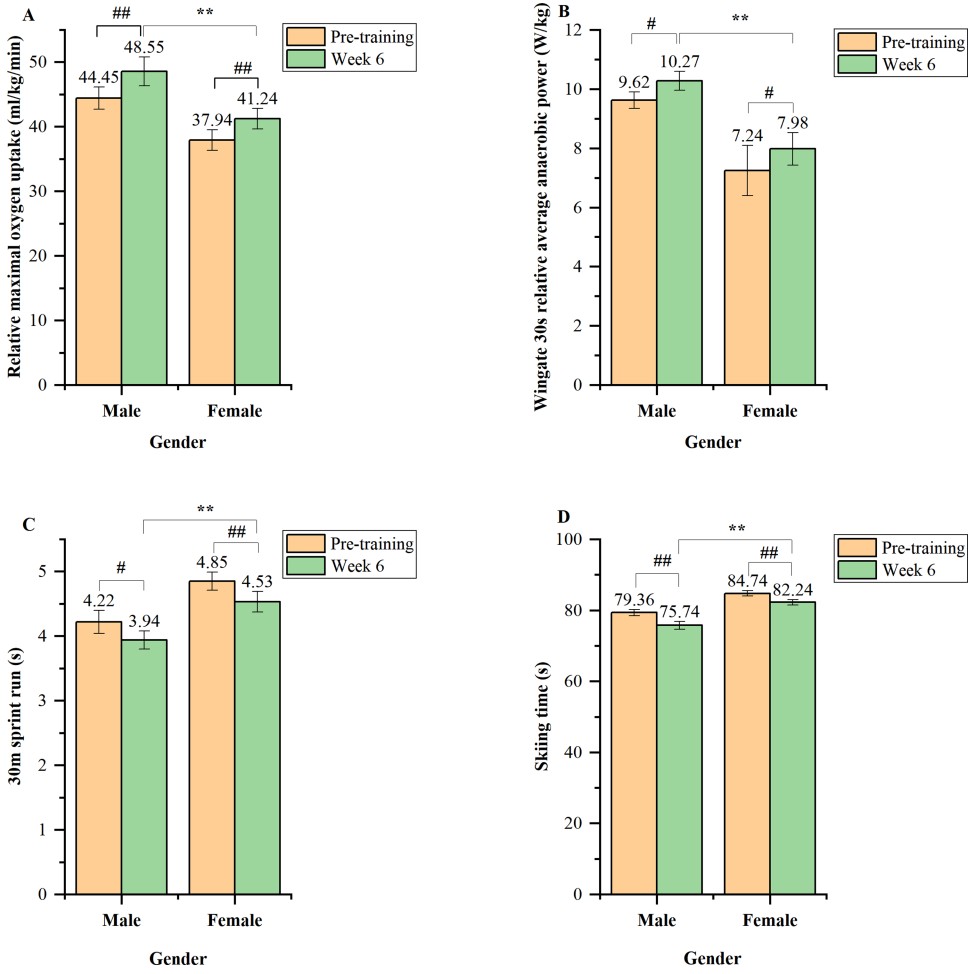

**Figure 1** **The changes in athletic ability involving relative maximal oxygen uptake (A), Wingate 30 s relative average anaerobic power (B), 30m sprint run performance (C), and race performance (D) after 6 weeks of sub-plateau cold environment training.** Notes: [#] Significant difference compared to pre-training ($p < 0.05$), [##] very significant difference compared to pre-training ($p < 0.01$), ** very significant difference compared to male ($p < 0.01$).

metabolism for energy supply and train and compete in cold alpine environments. There are significant differences in the physiological functions of the human body when exercising in altitude and cold environments compared to plains and ambient environments (*Acosta et al., 2018*; *Brajkovic, Ducharme & Frim, 1998*). Normally, when training at altitude, the body is more deeply stimulated than in a plain environment due to the relatively low oxygen concentration of the altitude environment (*Ramos-Campo et al., 2018*; *Schmutz et al., 2010*). RBC and Hb reflect the oxygen-carrying capacity of the blood and the functional status and their reserve levels in the body reflect aerobic metabolism. A number of studies have shown that plateau training can improve the aerobic metabolic capacity of athletes by increasing the density of skeletal muscle capillaries and the number of mitochondria, increasing blood volume and total Hb. This, in turn, enhances the ability of skeletal muscle

to buffer BLA, oxidize fatty acids, utilize energy efficiently during exercise (*Saunders et al., 2013*; *Tannheimer et al., 2010*). In this study, the RBC and Hb of both male and female athletes were higher than pre-training in the other weeks of the 6 weeks sub-plateau cold environment training, except for the female athletes whose RBC and Hb were slightly lower at week 6. This effect suggests that the 6 weeks of training promoted the production of RBC and increased the RBC reserve, thereby possibly contributing to the enhancement of maximum oxygen uptake. In addition, the changes in RBC and Hb of the athletes in this study indicate that the oxygen transfer capacity of the subjects improved in contrast to the pre-training level, which is consistent with other studies observing the increase of Hb level and oxygen-carrying capacity of the Hb after plateau training (*Bonne et al., 2014*; *Nummela et al., 2021*; *Wachsmuth et al., 2013*). Furthermore, this study found that both RBC and Hb were significantly lower in female athletes than in males throughout the 6 weeks of training in a sub-plateau cold environment, suggesting that there are significant gender differences in human RBC and Hb production. Lastly, this study also found that the RBC and Hb levels of athletes were significantly higher than those engaging in summer Olympic sports (*Ma, Gao & Li, 2019*). This may be due to the fact that the cold hypoxic environment of the plateau stimulated the body's erythropoietin (EPO) response (*Kasperska & Zembron-Lacny, 2020*; *Wisniewska, Ploszczyca & Czuba, 2020*).

### Effects of 6 weeks of sub-plateau cold environment training on the immune system of elite parallel giant slalom athletes

The acute exposure to high altitude typically causes an inflammatory response in the body and reduces immunity, while long-term altitude training may improve the stability of the immune system (*Aksel, Corbacioglu & Ozen, 2019*; *Gallagher & Hackett, 2004*; *Hackett & Roach, 2001*). Several studies have shown that short or intermittent plateau hypoxia exposure rapidly activated the sympatho-adrenal system to regulate the activity of immune cells and the secretion of related cytokines (*Jung, Kim & Park, 2020*; *Park et al., 2020*; *Svendsen, Hem & Gleeson, 2016*), while longer plateau training or hypoxia exposure may affect the immune system by regulating the proliferation and differentiation of immune cells *via* the hypoxia-inducible factor (HIF) signaling pathway (*Domingo-Gonzalez et al., 2017*; *Taylor & Colgan, 2017*; *Wang, Huang & Gao, 2021*). The proliferative capacity of T lymphocytes and the activity of NK cells are suppressed at the beginning of the hypoxia exposure (*Facco et al., 2005*; *Morabito et al., 2016*), the activity of NE cells is increased (*Chouker et al., 2005*; *Wang et al., 2019*), and the body experienced a certain inflammatory response. In contrast, after prolonged hypoxia and plateau training, the expression of CD55 and CD59 on the surface of leukocytes and CD4/CD8 of T lymphocytes increases, which in turn improves the regulatory function of the immune system (*Wang, Huang & Gao, 2021*). Theoretically, when training in a hypoxic environment on the plateau, the lack of oxygen combined with the training load causes the athlete enough stimulation. Once cold is added to the stimulation, the organism is subjected to a more serious burden, and the function of the organism's systems will undergo more complex changes. However, little research has been reported on these changes. In one study, however, the results demonstrated that when training in hypoxic and cold environments at plateau, the body may experience a series of

physiological stress responses similar to those seen when training at plateau or in hypoxic environments, such as a decrease in WBC and LY% reflecting immune function, and an increase in NE% reflecting inflammation during early training in cold environments on the plateau (*Thompson-Torgerson et al., 2007*). In our study, WBC and LY% were lower at week 2 of training in a sub-plateau hypoxic cold environment prior to the Olympic Games Test Competition compared to pre-training, while MO% and NE% were higher than pre-training, indicating that the athletes' immune system function was somewhat suppressed and the body experienced a certain inflammatory response during the first 2 weeks of training. Moreover, from week 4 to 6, the LY% level gradually increased and was higher than pre-training at the end of the sub-plateau hypoxic cold training, which was suggestive of recovery. On the other hand, WBC increased at week 4 and remained lower than pre-training at the end of the sub-plateau hypoxic cold training, while MO% and NE% gradually decreased from week 4 to the end of the training. This is generally consistent with *Gao et al. (2018)* and *Wang, Huang & Gao (2021)*, who reported the changes in indicators such as WBC, LY%, MO%, and NE% in rowers during hypoxic training. Therefore, the elite parallel giant slalom athletes' bodies adapted to the training load and the hypoxic cold environment of the sub-plateau, the stability of the regulatory functions of the body systems improved, immunity gradually recovered, and the inflammatory response was progressively reduced during the training period.

## Effects of 6 weeks of sub-plateau cold environment training on the musculature of elite parallel giant slalom athletes

Previous studies have found that the human body becomes less adapted to training at the early stage of sub-plateau training, as indicated by increased protein and amino acid catabolism and BU and CK levels (*Wang, Gao & Gao, 2013*; *Yu et al., 2016*). As the body adapts to the plateau training environment, the BU and CK levels gradually stabilize and change regularly with the training volume or intensity (*Li & Wang, 2017*; *Zhang, Gao & Zhu, 2017*). In our study, the CK for both genders and BU for male were lower at week 2 compared to pre-training. This is consistent with *Wang, Gao & Gao (2013)*, who reported the same change in blood BU and CK levels in rowers when training at plateau for longer than 1 week. As for this phenomenon, this may be related to the adaptation of the athletes' body to the hypoxic environment of the plateau and the improved ability of the body to withstand the training load. In addition, the BU and CK levels of both male and female in this study were significantly higher at the end of week 6 compared to pre-training and week 4, which may be related to greater stimulation of the athlete's body caused by the 6-week multiple competition-intensity specific skiing technique training. Further analysis revealed that although elite parallel giant slalom athletes' BU and CK were significantly higher at the end of week 6 of training, they were still in the normal range. BU and CK did not significantly exceed the normal range during the last 2 weeks of high intensity training, and the body remained stable in terms of the functional status and muscle fibers. This indicates that training adaptation ability of the athletes' body also improved during this process, though the stimulation of the training load on the body gradually deepened with the increase of the training load. Previous studies have shown that prolonged hypoxic

exposure can promote HIF-1 gene expression and induce adaptive changes in the body to the hypoxic environment and improve exercise performance (*Cimino et al., 2012*; *Rocco et al., 2014*; *Yeo, 2019*). As a result, prolonged and repeated cold stimulation can promote adaptation through adrenergic and non-adrenergic mechanisms and reduce the physiological fluctuations during exercise in cold environments (*Castellani & Young, 2016*; *Saltykova, 2016*; *Young & Castellani, 2007*).

## Effects of 6 weeks of sub-plateau cold environment training on the endocrine system of elite parallel giant slalom athletes

T has the function of promoting anabolism in the body, inhibiting the breakdown of muscle glycogen and activating glycogen synthesis, increasing muscle glycogen and creatine phosphate reserves, and enhancing immunity and resistance to bacterial infection (*Chiu et al., 2015*; *Ouergui et al., 2016*). In contrast, the main role of C is to accelerate the breakdown of fats and proteins into sugars, and accelerate catabolism (*Ouergui et al., 2016*; *Pesce et al., 2015*). T and C reflect anabolism and catabolism from different perspectives, and can be used to assess the effect of training load on an athlete's functional status. The training loads in different environments have unique effects on the body. When exercising in hypoxic or cold environments on the plateau, the athletes were stimulated by the training load and experienced the hypoxia and cold stimuli (*Ramos-Campo et al., 2018*; *Schmutz et al., 2010*). *Wang (2020)* showed that elite modern pentathletes had a high anabolic capacity and peak serum T levels during the sub-plateau adaptive training phase of moderate intensity training, while a certain decrease in serum T occurred during the increased sub-plateau training load phase. These results suggested that changes in serum T in athletes were associated with training load during the subalpine period. *Zhao, Li & Lu (2008)* showed that the serum T and C of male weightlifters decreased significantly at the beginning of sub-plateau training compared to the pre-training period, increased slightly at the middle of training period, and decreased slightly at the end of sub-plateau training. Serum T and C were lower throughout the sub-plateau training period compared to the pre-training period. *Zhao, Li & Lu (2008)* concluded that this was related to the greater maladjustment of the body to the training load and plateau environment during sub-plateau training in male weightlifters. In our study, serum T and T/C levels were significantly lower in both males and females during the first 2 weeks of training in a sub-plateau cold environment compared to the pre-training period, while serum C was significantly higher compared to the pre-training period. These results suggested that the training load at the beginning of training, combined with the hypoxic and cold plateau environment, caused some pressure to the participants' body. From weeks 4 to 6, the athletes' serum T and T/C gradually increased rather than decreasing due to the increase in training load. These results suggest that as the training progressed, the body systems gradually adapted to the sub-plateau cold environment. Moreover, the anabolic levels of the elite parallel giant slalom athletes improved and the physical function status was higher after 6 weeks of training in a sub-plateau cold environment.

### Effects of 6 weeks of sub-plateau cold training on the aerobic metabolic capacity of elite parallel giant slalom athletes

The existing studies concerning plateau or hypoxic training have demonstrated that plateau or hypoxic training lasting 3 weeks or more could increase the RBC and Hb counts of the athletes, improve oxygen transport capacity, enhance the removal of BLA, thereby improving aerobic metabolic capacity (*Bonne et al., 2014*; *Saunders et al., 2013*; *Shaw et al., 2020*; *Tannheimer et al., 2010*; *Wachsmuth et al., 2013*). It has been shown that plateau or hypoxic training not only improves indicators related to oxygen transport and oxygen utilization capacity (*e.g.*, EPO, Hb, RBC, 2, 3-DPG) but also promotes maximal oxygen uptake and specific aerobic athletic ability (*Gao et al., 2018*; *Liu & Zhang, 2015*; *Wang, Gao & Gao, 2013*). Our results showed that 6 weeks of training in a sub-plateau cold environment resulted in a significant increase in the relative maximum oxygen uptake levels in elite parallel slalom athletes. The data suggested that 6 weeks of specialized ski technique training and physical training in a sub-plateau cold environment may improve aerobic metabolism capacity. This trend was also consistent with the changes of RBC and Hb levels. RBC and Hb levels remained elevated and stable after training, and there was a significant increase in maximum oxygen uptake. This was also consistent with the findings of *Wang, Gao & Gao (2013)* that sub-plateau training could improve rowers' RBC and Hb levels and improve their aerobic metabolism capacity. Therefore, it could be concluded that this training was effective in improving the aerobic metabolism capacity of athletes in snowboarding parallel giant slalom.

### Effects of 6 weeks of sub-plateau cold environment training on anaerobic metabolic capacity in elite parallel giant slalom athletes

Some studies have shown that plateau or hypoxic training enhances aerobic metabolism and promotes anaerobic metabolism (*Bădău et al., 2016*; *Christoulas, Karamouzis & Mandroukas, 2011*; *Holmberg, 2015*). Hypoxia can induce the expression of HIF-1 and HIF-1-related genes (*Cimino et al., 2012*; *Yeo, 2019*). As a DNA-binding protein, HIF-1 induces an increase in the expression of several genes, including glycolytic metabolism enzymes, which promotes adaptation to hypoxic environments (*Tang & Jiang, 2004*). *Semenza & Wang (1992)* showed that both EPO-synthesizing Hep3B cells and non-EPO-synthesizing HeLa cells were induced by hypoxia to increase mRNA transcription of three key enzymes of glycolytic metabolism, aldolase A (AL-DA), phosphofructokinase (PFK), and pyruvate kinase (PK). Therefore, it can be suggested that hypoxia may promote an increase in glycolytic metabolic enzymes induced by hypoxia *via* the induction of HIF-1 expression. *Wang (2013)* and *Li (2014)* showed that hypoxic training stimulated the body more deeply and improved BLA levels of athletes compared to training in a normoxic environment. Their studies confirmed that 3 to 4 weeks of hypoxic training can improve anaerobic levels in boxing and sparring athletes who are mainly energized by anaerobic glycolytic metabolism. This may improve the specific athletic ability of these athletes. In addition, some studies have shown that 4 weeks of hypoxic training can significantly improve the levels of all indicators of the Wingate 30 s anaerobic work test in aerobic and cycling athletes, and can promote the reduction of BLA and CK levels (*Li, 2020*; *Ma et al.,*

*2013*). In our study, subjects had significant improvements in anaerobic exercise capacity such as Wingate 30 s anaerobic power levels and 30 m sprint runs, as well as significant improvements in competition performance. These results indicate that the 6-week training was beneficial to improving anaerobic metabolism and increase competitive performance.

### Limitations

The aim of this study was to investigate the effects of six weeks of training in a sub-alpine cold environment on the physical functioning and athletic ability of elite snowboarding parallel slalom athletes. This study showed that elite national snowboarders had significant improvements in aerobic and anaerobic capacity and race performance after six weeks of medium intensity physical training and high intensity ski-specific technical training in a sub-alpine cold environment. The small sample size of this study (only nine elite national team snowboarding parallel giant slalom athletes) is a limitation of this study that may impact the generalizability of the results. However, as there are only nine athletes in the entire snowboarding parallel giant slalom national team, these athletes are some of the best snowboarding parallel giant slalom athletes in China. Therefore, this study may provide some reference value for the national team in the future in terms of training methods, even for athletes of other competitive levels and other winter snow sports. Of course, in order to improve the applicability of the training methods in this study and to provide a greater reference value for other sports, we believe that the sample size of the study should be expanded. It would be of value to conduct a randomized controlled trial based on this (*e.g.*, both the test group and the control group use the same training arrangement, with the test group training in a cold sub-alpine environment and the control group training in a plain environment) to investigate the extent that the content and the training environment improved athletic performance, respectively.

## CONCLUSIONS

Six weeks of specialized high intensity ski technique training and medium-to-high intensity physical training in a sub-plateau hypoxic cold environment may improve the physical function, and the aerobic and anaerobic capacity of elite snowboarding parallel giant slalom athletes. A decrease in immunity and anabolism appeared in the early stages of the training sessions. Furthermore, male athletes showed a greater improvement in physical functioning and athletic ability when trained in sub-plateau cold environments.

## ACKNOWLEDGEMENTS

We are very grateful to the national Chinese snowboarding parallel giant slalom athletes for their participation in this study. In addition, we are very grateful to the coaches and managers of the national Chinese snowboarding parallel giant slalom sports team for their support of this study.

### Funding

This work was supported by the National Key Research and Development Program of Science and Technology for Winter Olympics Key Project (NO. 2019YFF0301603), the National Snowboarding Parallel Giant Slalom Team Preparation for the Winter Olympics Research and Scientific Services (NO. TC210Q0LP) and the Shanghai Key Laboratory of Human Movement Development and Protection (Shanghai University of Sports) funded project (11DZ2261100). The funders had no role in study design, data collection and analysis, decision to publish, or preparation of the manuscript.

### Grant Disclosures

The following grant information was disclosed by the authors:
National Key Research and Development Program of Science and Technology for Winter Olympics Key Project: 2019YFF0301603.
National Snowboarding Parallel Giant Slalom Team Preparation for the Winter Olympics Research and Scientific Services: TC210Q0LP.
Shanghai Key Laboratory of Human Movement Development and Protection (Shanghai University of Sports) funded project: 11DZ2261100.

### Competing Interests

The authors declare there are no competing interests.

### Author Contributions

- Tao Ma conceived and designed the experiments, performed the experiments, analyzed the data, prepared figures and/or tables, authored or reviewed drafts of the article, and approved the final draft.
- Jingwang Tan performed the experiments, authored or reviewed drafts of the article, and approved the final draft.
- Ran Li performed the experiments, authored or reviewed drafts of the article, and approved the final draft.
- Jiatao Li performed the experiments, authored or reviewed drafts of the article, and approved the final draft.
- Binghong Gao conceived and designed the experiments, performed the experiments, authored or reviewed drafts of the article, and approved the final draft.

### Human Ethics

The following information was supplied relating to ethical approvals (i.e., approving body and any reference numbers):

The study was conducted in accordance with the Declaration of Helsinki and approved by the Ethics Committee of Shanghai University of Sport (Ethics number: 102772020127082; Date of approval: 27 October 2020).

## Data Availability

   The raw measurements are available in the Supplemental Files.

## Supplemental Information

Supplemental information for this article can be found online at http://dx.doi.org/10.7717/peerj.14770#supplemental-information.

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
