# Peer review of "Effects of six weeks of sub-plateau cold environment training on physical functioning and athletic ability in elite parallel giant slalom snowboard athletes"

_PeerJ, doi:10.7717/peerj.14770_

## Round 0.1 · original submission · Minor Revisions

The article has merit, and the reviewers agreed on that. However, some methodological aspects can be better described.

·

Basic reporting

The manuscript “Effects of 6 weeks sub-plateau cold environment training on physical functional status and athletic ability in elite snowboarding parallel giant slalom athletes” has the main objective of to investigate the effects of 6 weeks of training in a sub-plateau cold environment on physical functional status and athletic ability among elite snowboarding parallel giant slalom athletes. This is an original and interesting manuscript, i suggest minor revisions so that the manuscript can be considered for publication in PeerJ.

Experimental design

ABSTRACT: I suggest replacing "Objective" with "Background".
INTRODUCTION:
The manuscript introduction is well written and can guide the reader to the problem and research objective.

MATERIALS AND METHODS:
Training arrangements
Lines 124 – 134 – I suggest indicating how the intensity of the training sessions was controlled. Furthermore, I suggest that the authors clarify in this topic and in Table 2, or by including supplementary material, the trainings that were carried out in the different preparation periods. So that future readers of the manuscript can understand and replicate the methodology.

30 m sprint run test
Lines 148 – 154 – I suggest that the authors indicate the technical specifications and characteristics of the equipment that was used to measure the 30-meter sprint.


Wingate 30-second anaerobic athletic ability test
Lines 157 – 162 - I suggest that the authors indicate a reference to the protocol used.

Maximal oxygen uptake assessment
Lines 165 – 161 - I suggest that the authors indicate the characteristics of the equipment used to measure the athletes' oxygen consumption. In addition, it is important to explain the criteria considered for the evaluations to be closed. Finally, it is important that a reference is presented to support the evaluation protocol used.

Race performance
Lines 174 – 179 - I suggest that the authors indicate how the time in the test was monitored. In addition, it is important to indicate the characteristics of the equipment used to carry out the tests.

Statistical analysis
Lines 182 – 193 - I suggest that the authors indicate a reference for the method of analysis of the effect size used, in addition, that the qualitative analysis of the different magnitudes of difference be indicated. It is important to indicate whether any of the variables presented a non-parametric distribution.

Validity of the findings

RESULTS
Tables and figures must be self-explanatory. I suggest that the authors indicate in the notes or in the legends of the tables and figures the meaning of the abbreviations used.

DISCUSSION
Congratulations, the discussion is very clear and with good references, discussing with the scientific literature the main results of the study.

I suggest that at the end of the discussion, the authors indicate the limitations of the study and suggestions for future studies. Also, due to the small sample size, do the authors believe that these results can be extrapolated or is this a limitation of the study? Finally, I suggest that the authors present more clearly the main practical applications of the results found, in both sexes, so that coaches and athletes can use this specific information in their training.

Additional comments

I suggest that authors review PeerJ formatting guidelines to suit the manuscript.

·

Basic reporting

Blood index test and procedure
Please add the protocol reference used.

Wingate 30-second anaerobic athletic ability test
Please add the protocol reference used.

Maximal oxygen uptake assessment
Please add the protocol reference used.

Experimental design

MATERIALS AND METHODS
Participants and ethical principles
Please add inclusion and exclusion criteria. Could the athletes be injured? Should athletes participate in a minimum of training sessions?

Training arrangements
As the study presents the result of an intervention added to the effect of the environment, it is important that the training be described in more detail. I think that the practical application of the study lies in the understanding of the intervention in these conditions, since it is not possible to identify whether it was the environment, the training or the combination of the two. How many workouts could the athlete miss? Please make this clear.


Maximal oxygen uptake assessment
Please add which rotation per minute was used or velocity

Validity of the findings

Discussion
I suggest including the limitations and positives at the end. I think it should be clear to the reader the limitations of a low sample number, even if the sample is of good quality.

Additional comments

The manuscript aimed to the effects of 6 weeks of training in a sub-plateau cold environment on physical functional status and athletic ability among elite snowboarding parallel giant slalom athletes. The manuscript is well written and describes the mechanisms that explain the improvement with training on the imposed conditions. My main concern is with the methods session. I suggest improving this session so that the study can be applied by coaches. In particular, training should be better described. Thank you for the opportunity to review this manuscript. Congratulations on the work.

---

## Round 0.2 · accepted · Accept

After confirmation from both reviewers, the article is ready to be accepted.

·

Basic reporting

I congratulate the authors for peer-reviewing the suggestions made by the reviewers. The manuscript is very relevant and should be accepted for publication in PeerJ.

Best regards,

Artur Preissler

Experimental design

no comment

Validity of the findings

no comment

Additional comments

no comment

·

Basic reporting

no comments

Experimental design

no comments

Validity of the findings

no comments

Additional comments

Dear editor,

The manuscript is relevant for publication. The authors reviewed point by point. I consider that the article can be accepted for publication.

Thank you for the opportunity to review the manuscript.

Best regards,

Pedro Schons